# The Evaluation of Energy Availability and Dietary Nutrient Intake of Sport Climbers at Different Climbing Levels

**DOI:** 10.3390/ijerph20065176

**Published:** 2023-03-15

**Authors:** Anna Chmielewska, Bożena Regulska-Ilow

**Affiliations:** Department of Dietetics and Bromatology, Wroclaw Medical University, 50-367 Wroclaw, Poland

**Keywords:** sport climbing, diet, energy availability

## Abstract

Proper nutrition is the basis for athletes’ performances when competing or training. The increasing training volume accompanying the increase in advancement should go hand in hand with the appropriate supply of energy as well as macro and micronutrients. The diet of climbing representatives due to the desire to achieve a low body weight may be deficient in energy and micronutrients. Our study aimed to evaluate the differences in energy availability and nutrient intake of female and male sport climbers at different climbing levels. Anthropometric parameters and the resting metabolic rate were measured, the questionnaire about climbing grade and training hours was filled, and a 3-day food diary was fulfilled by 106 sport climbers. Based on the collected data, the energy availability as well as the macro- and micronutrient intake was calculated. Low energy availability (EA) was observed among both genders of sport climbing representatives. A significant difference between EA in various levels of advancement was found in the male group (*p* < 0.001). Differences in carbohydrate intake (g/kg/BW) between sexes were observed (*p* = 0.01). Differences in nutrients intake between climbing grade were found in both the female and male groups. In the group of female elite athletes, the adequate supply of most of the micronutrients can imply a high-quality diet despite the low calorie content. It is necessary to educate sport climbing representatives about the importance of proper nutrition as well as the consequences of insufficient energy intake.

## 1. Introduction

Sport climbing is rapidly growing in popularity, and according to the International Sport Climbing Federation, approximately 25 million people practice it globally. In the United States alone, as many as 1000–1500 new people try their hand at the sport every day. Furthermore, it is estimated that approximately 100,000 people practice the sport regularly in over 150 indoor climbing facilities in Poland [1].

Sport climbing was included in the Olympics for the first time in 2021 and comprised three disciplines, namely bouldering, speed climbing, and lead climbing [2]. Bouldering and lead climbing are the most frequently practiced disciplines [3]. Bouldering is performed rope-less with spotters and crash mats for protection, and indoor walls typically do not exceed four meters in height [4]. Lead climbing is a discipline based on leading long routes (usually 20–40 m) using a rope and quickdraws clipped to bolts placed in the rock or an artificial wall [5].

Characterized by various dynamic movements under conditions of compensated fatigue and changes in work intensity, sport climbing is considered a complex sport [6]. Climbing requires a significant proportion of whole-body aerobic capacity, with the anaerobic power more important for challenging routes with steeper angles [7]. All types of climbing require a high level of technique, strength, and endurance. However, during longer climbing routes, stamina is emphasized, whereas bouldering relies on power [8].

Sport climbers present with specific anthropometric characteristics similar to the profiles of ballet dancers and long-distance runners. Climbers are typically shorter, leaner, and lighter than non-climbing athletes, and elite climbers are short in stature and have low body mass and body fat [3]. A high power-to-mass ratio (SMR) is considered advantageous in rock climbing, and losing weight to tackle more challenging routes is reported among sports climbers [3]. However, consuming a diet too low in energy can place climbers at risk of the inadequate intake of nutrients such as carbohydrates (CHOs), protein, calcium, and iron. It can also cause fatigue and a weakened immune system [6]. As such, these behaviors put climbers at risk of restrictive eating and increase the potential for eating disorders (ED), disordered eating (DE), laxative/diuretic use, and relative energy deficiency in sport (RED-S) [3].

Proper nutrition is the basis for an athlete’s performance when competing or training. Indeed, a well-balanced diet provides energy when athletes perform various physical activities and is also necessary for post-match recovery [9].

An increasing number of studies have investigated the aspects of factors that contribute to a successful climbing performance, such as anthropometric parameters, bio-mechanics, and physiological and psychological factors [10]. However, the number of studies evaluating energy intake (EI) and assessing the dietary intake of sports climbers is still limited. Some studies involving small groups of climbers from different countries analyzed the intake of energy, macronutrients, and micronutrients [11,12,13], diet composition [14,15,16], and antioxidant intake [17].

Some considerations have been given to the nutritional requirements for Olympic-style climbing [3] and bouldering [4].

Thus far, research has found inadequate energy availability (EA) [7], unbalanced food quality, and poor nutrient timing in the diets of climbers of different ages and levels of advancement [18]. Furthermore, several studies have investigated supplement use in sports climbers [12,19,20] and analyzed supplement protocol use [21,22].

However, most of those studies involved a small group of top-level climbers, competitive representatives, training in one sport object. Our study aimed to involve a greater number of climbers at different levels of advancement, training on numerous sport facilities, which gives a bigger possibility to observe the differences in the general population of climbers of various grades.

This study aimed to assess the dietary nutritional intake of sports climbers, investigate whether it matches the recommendations, and understand the main dietary deficiencies that occur.

Along with the level of advancement of sports climbing and the growing number of training methods, the focus should shift to different factors that may influence the performance, including proper nutrition. It was assumed that differences would exist between the dietary nutrient intake of climbers of different levels, with a tendency for elite climbers to have a better nutrition pattern that included higher vitamin and mineral components. On the other hand, it was assumed, as in previous studies, that sport climbers tend to lower their EI to obtain a low body mass. This can be a reason for malnutrition and insufficient EA.

This study aimed to evaluate the differences in the EA and the macronutrient and micronutrient intake of sports climbers, females and males, at various climbing levels.

## 2. Materials and Methods

### 2.1. Design

Study information was spread through social media and directly through the trainers of the five climbing gyms active at the time. The inclusion criterion for the study was at least one year of regular sports climbing. The exclusion criterion was recreational climbing with practice less than once a week. All climbers who were willing to take part for the duration of the study and who fulfilled the criteria were accepted. Data collection took place between June 2019 and August 2021 with the obligatory break during the coronavirus pandemic between January 2020 and February 2021. As the level of advancement of participants was set based on the climbing routes crossing from the previous 6 months, the study performance was refrained for a longer period after the opening of sports facilities and the end of the strict quarantine, so the data were reliable. The Institutional Ethics Committee of Wroclaw Medical University approved the study (number KB-45/2019), which was conducted in accordance with the Declaration of Helsinki.

### 2.2. Procedure

The procedures of the indirect calorimetry test, anthropometric measurements, and filling the questionnaire were carried out in a special design area in the Department of Dietetics, with the assistance of a trained nutritionist. It was requested that the study participants attend after fasting, an overnight rest, and having avoided of intense physical activity in the evening before the measurements were taken. The study participants were asked to come during the morning hours.

Resting heart rate, systolic blood pressure, and diastolic blood pressure were measured using a sphygmomanometer to exclude the performance of indirect calorimetry in a state of anxiety. Anthropometric measurements, including body weight (BW), body fat, and fat-free mass (FFM), were performed with the X-CONTACT 356 analyzer (Jawon Medical Co., Seoul, Korea), and height was measured using a TANITA HR-001 stadiometer (TANITA, Tokyo, Japan). The energy needs of the study participants were determined by measuring their resting metabolic rate (RMR) with indirect calorimetry (IC) using the Fitmate WM (Cosmed, Rome, Italy) device. It was requested that the study participants attend after fasting, an overnight rest, and the avoidance of intense physical activity in the evening before the measurements were taken. The test took place in a darkened and soundproof room and lasted for 12 min. Participants’ breathing patterns were matched to the measurement requirements in the first two minutes, with the RMR measurement taking ten minutes. Contraction force measurement used a MAP 80K1 hand grip dynamometer (KERN & SOHN GmbH, Balingen, Germany). The contraction force of each hand was measured twice, and the mean value of two measurements was calculated.

After performing the measurements, participants filled out the questionnaire. Each participant’s climbing grade was subjectively determined by assessing three different routes established over the preceding six months. Participants were asked to indicate the highest grade, lead, or boulder that they had managed to redpoint on three different routes/problems on either an artificial wall or a rock. Participants also declared their frequency and time of training in the week’s perspective.

Next, each participant was asked to deliver a three-day food diary using the Fitatu mobile application [23]. Participants were instructed to enter each meal in the application, as it gives the possibility to convert the house measures of food products for grams. The application allows to scan products based on barcodes and contains information about suggested product portions, which makes it easier to fill in the diary. Photographs of the meals consumed supplemented the data and provided accurate information on portion sizes. In case of any doubts according to product and meal types and sizes, the participants were contacted to specify the information.

### 2.3. Participants

One hundred and fourteen regular sports climbers, who were willing to take a part in the study and fulfilled the criteria, which were from Wroclaw in Poland, took part in the study. Of the study participants, 106 (40 females and 66 males) delivered dietary reports suitable for analysis.

### 2.4. Outcome Measures

Based on the indirect calorimetry, blood pressure, and anthropometric and contraction force measures, the characteristics of the group were established. BMI value was calculated using body mass measured with the bioimpedance balance and height measured with a stadiometer mentioned in the design part. For the grip strength-to-body mass ratio (SMR), the mean value of all contraction force measures was used.

The grade declared in the questionnaire was standardized according to the International Rock Climbing Research Association (IRCRA) scale [24]. For female climbers, an Intermediate score was between 10 and 14, an advanced score was between 15 and 20, and an elite score was between 21 and 26. For male climbers, an intermediate score was between 10 and 17, an advanced score was between 18 and 23, and an elite score was between 24 and 27.

To assess the energy availability, the measurement of the energy exercise expenditure (EEE) was calculated based on the training time declared in the questionnaire and the activity logs using the metabolic equivalent of task (MET) [25]. Based on the obtained calculations and fat-free mass (FFM) value obtained with the measurements with the bioimpedance scale, the EA was calculated.

EA was calculated using the following equation [26]:$$\mathrm{EA}\;=\frac{\mathrm{EI}\;–\;\mathrm{EEE}}{\mathrm{FFM}}$$

To assess the macro- and micronutrients intake, the food records were transferred to ESHA’s Food Processor^®^ Nutrition Analysis software 11.7.217 database 11.7.1 (ESHA Research, Salem, OR, USA), which contains the Nutrient Tables of Foods database [27]. For the macronutrient intake analysis, the dietary macronutrient intake was converted to g/kg/BW as the general recommendations for athletes are expressed as g/kg/BW. Dietary intake was compared to Norms for the Polish Population from 2020 [28] for the female and male groups separately, as the nutritional recommendations are gender-specific.

#### Statistical Analysis

Considering the small sample size, and therefore the small power of a normality test, the non-parametric tests were used. The differences in the dietary intake of macronutrients and micronutrients and EA between the three climber groups (intermediate, advanced, and elite) were assessed using the Kruskal–Wallis test, and the differences between genders were assessed using the Mann–Whitney U test. Differences were recognized as statistically significant when *p* < 0.05. For statistically significant differences, a post hoc Dunn test with Bonferroni correction assessed the differences between the study groups. Descriptive statistics were generated for the anthropometric measurements of the study group and the dietary intake of macronutrients and micronutrients. Data analysis was performed with STATISTICA 13.1 (StatSoft Inc., Tulsa, OK, USA).

## 3. Results

The characteristics of the study groups based on the gender and climbing level are presented in Table 1. Significant differences were observed in systolic (*p* = 0.03 for elite vs. intermediate; *p* = 0.01 for elite vs. advanced) and diastolic (*p* = 0.03 for elite vs. intermediate) blood pressure in female climbing level groups. In males, differences in SMR (*p* < 0.01 for elite vs. intermediate; *p* = 0.03 for elite vs. advanced), contraction force of the left hand (*p* = 0.03 for elite vs. intermediate), and body mass index (BMI) (*p* = 0.04 for elite vs. intermediate) existed between climbing levels.

Table 2 presents EA in the groups of male and female climbers of different climbing levels. EA decreased with a higher climbing level in both groups, which was significant in males but non-significant in females. Among the males, there were significant differences between the post hoc tests of the elite vs. intermediate groups (*p* = 0.04; Z = 2.47) and advanced vs. intermediate groups (*p* = 0.01; Z = 2.86). Moreover, EA was significantly different between the genders. Comparing the genders in each climbing level revealed significant differences in the advanced group (*p* = 0.04; Z = 2.04) with no significant differences between the intermediate (*p* = 0.12; Z = 1.54) and elite (*p* = 0.8; Z = 0.21) groups.

Suboptimal EA (>30 kcal/kg FFM/day) was reached in 73% of the intermediate, 45% of the advanced, and 40% of the elite female group. In male groups, only 47% of intermediate participants and 21% of the advanced group reached a suboptimal EA level. None of the elite male climbers exceeded the value of 30 kcal/kg/FFM. In the intermediate and advanced level, median EA values in male groups were lower than in female groups.

The dietary intake of macronutrients in male and female groups on different climbing levels is presented in Table 3. No significant differences were found between climbing levels, although differences were found in the CHO intake between the male and female cohort. However, when measuring differences in the CHO intake between genders in each climbing level, marginally significant differences were found in the intermediate (*p* = 0.06; Z = −1.88) and the elite (*p* = 0.06; Z = −1.88) group, with no significance in the advanced group (*p* = 0.13; Z = −1.50)

The dietary intake of macro and micronutrients of male and female climbers on different climbing levels is presented in Table 4. In females, the differences were observed in fiber (*p* = 0.007; Z = 3.06), iron (*p* = 0.003; Z = 3.33), magnesium (*p* = 0.008; Z = 2.99), phosphorus (*p* = 0.04; Z = 2.46), and potassium (*p*= 0.05; Z = 2.43) intake between the elite and advanced groups. Difference in iodine (*p* = 0.01; Z = 2.89) intake was found between advanced and intermediate groups. According to zinc intake in the post hoc test, no significant differences were confirmed.

Males in the advanced and intermediate groups differed significantly in vitamin B6 (*p* = 0.4; Z = 2.50), phosphorus (*p* = 0.02; Z = 2.72), iron (*p* = 0.3; Z = 2.60), and zinc (*p* = 0.002; Z = 3.42) intake. According in the post hoc test, no significant differences in potassium intake were confirmed.

Compared to the estimated average requirement (EAR) and adequate intake (AI) values for vitamin D, vitamin A, sodium, and potassium for the Polish population, the proper intake was reported for most micronutrients in all study groups. However, all groups reported an insufficient intake of vitamin D and iodine. In the advanced and intermediate female group, insufficient amounts of calcium and potassium were also reported. The elite group of female climbers presented the lowest median EI, with the median value intake of most of the micronutrients fulfilling the recommendations. Elite and advanced males reported inadequate intakes of potassium.

## 4. Discussion

The study results partly confirmed the hypothesis on the differences between diet quality between the participants with different levels of advancement. Indeed, elite female climbers consumed significantly higher amounts of several nutrients than lower-level climbers. However, the male climber groups did not represent such a characteristic. None of the elite male group reached the suboptimal EA. However, the lowest median value of EA was reported in the elite female group.

The anthropometric measurements of body mass and height for both sexes were similar to the participants of other studies involving sports climbers [11,12,15,16]. However, the body fat percentage was higher in the male groups [12,16].

Climbers from the elite groups had the lowest BMI values, which were 20.1 in females and 21.2 in males. In a study by Sas-Nowosielski et al. [11], the BMI of climbers significantly predicted their climbing abilities on the most challenging route climbed in the so-called redpoint style. Kemmler et al. [16] demonstrated a moderate negative effect of low BMI value on bone mineral density (BMD). Nonetheless, climbers had higher BDM values in most studied regions compared to non-climbing controls with the same BMI.

Studies investigating EI in adolescents [29] and adult sports climbers [11,16,18] to date have reported the caloric value of the assessed diets to be too low for the predicted energy expenditure. Furthermore, few studies have focused on whether the dietary caloric intake provided enough EA in relation to the training volume.

EA is the dietary EI minus the energy expended during exercise, with EA being an input to the body’s physiological systems that remain after exercise training [18]. Low EA occurs when either the dietary EI is too low or the energy expanded through exercise is too high, and the energy needed for the maintenance of physiological functions such as bone health, the menstrual cycle, and metabolic and immune function becomes insufficient. There is no optimal EA for high-performance athletes. To date, in studies of sedentary normal-weight females, 45 kcal/kg FFM/ day was defined as a value for achieving optimal energy balance. Meanwhile, a study on men reported that 40  kcal/kg FFM/ day was enough to support energy balance. An EA of 30–45  kcal/kg FFM/ day is already considered a reduced EA, and athletes should only stay within this value for a short period, such as when aiming to reduce BW [30].

The analysis of the entire cohort in the current study showed that 36% presented with suboptimal EA (<45 kcal/kg FFM/day) and 59% had low EA (LEA; <30 kcal/kg FFM/day). LEA was demonstrated by 45% of female participants and 68% of male climbers. Median EA was the lowest in the female elite group (13.4). In male groups, there was a significant difference in EA between the elite and intermediate groups and between the advanced and intermediate groups. In the study by Monedero et al. [13], suboptimal EA and LEA were evident in 88% and 28% of climbers, respectively. The prevalence of suboptimal EA was 93% in male subjects and 82% in female subjects, and the prevalence of LEA was 29% in the male subjects and 27% in the female subjects.

In a study by Simič et al. assessing EA in adolescent climbers [7], the mean EA (27.5 ± 9.8 kcal/kg FFM/day) was below the recommended level, with no participant meeting the target of 45 kcal/kg FFM/day and 63% being in the range of LEA. Furthermore, 26% of the climbers failed to meet their predicted basal metabolic rate, and a significant difference in climbing levels was reported between the groups with suboptimal and low EA. Moreover, Michael et al. [29] demonstrated that 82% of adolescent climbers did not meet the recommended EI.

Monedero et al. [13] observed energy intakes that were significantly lower than daily requirements, but only in females. In a study by Gibson-Smith [12], the average EA was 41.4 ± 9 kcal/kg FFM/day, with a significantly higher EA in females than in males (45.6 ± 7 kcal/kg FFM/day vs. 37.2 ± 9 kcal/kg FFM/day, respectively). Furthermore, 78% of the elite adult climbers failed to meet the predicted energy required to support a moderate level of physical activity of 12 h of training per week, while 18% failed to meet the predicted RMR values [12]. In the current study, only the female intermediate group had a suboptimal EA, with a median EA <30 kcal/kg/FFM/day in the other groups. The median EA values were lowest in the elite groups of males and females. The median EIs in the elite female group was lower compared to the elite male group.

Those climbing at the highest levels may be involved in competition or in maintaining/acquiring sponsorships. These factors may lead to additional pressure to achieve a lower BW or leanness, which could negatively affect eating patterns [29]. Furthermore, a study by Modaberi et al. [31] reported a positive correlation between anxiety, emotional eating, and external eating behavior in top rope climbers. Previous studies analyzing DE reported higher prevalence rates among elite athletes [32]. However, in the studies involving a range of disciplines [33] and those focusing on adolescents [29] and adults [30], females were considered to be at the greatest risk of DE.

The results of our study show that the male groups presented a lower median of EAs in intermediate and advanced groups, however, the median of EAs in elite groups was smaller in female group. This is consistent with the results of previous research, which indicated that female groups presented with low EA and DE more often. Indeed, Joubert et al. [30] reported a DE prevalence of 6.3% in males and 16.5% in female climbers. Almost half of the females from the elite and high elite groups presented with DE (42.9%), and DE was only significantly associated with climbing ability in the female group. In Sas-Nowosielski et al., relatively more females (8 out of 10) than males (5 out of 13) were dissatisfied with their body mass and felt a need to slim down [11].

Female athletes under-eat for reasons unrelated to sport, with Wardle et al. [34] reporting that there are approximately about twice as many young women than men at every decile of BMI perceive themselves to be overweight. Furthermore, 25% of male athletes from esthetic, leanness-focused, or weight-sensitive sports show disordered dietary patterns, which was also closely associated with higher body fat percentages and body dissatisfaction [35].

In the context of exercise-related health risks, LEA has been extensively described in female athletes. However, the International Olympic Committee recently expanded the concept of the sportsmen triad to include the term relative energy deficiency in sports to describe the consequences of LEA on health and performance in both males and females [36].

It is suggested that an EA below 30 kcal/kg FFM/ day can lead to an abrupt decline in bone mineralization, with similar reductions in insulin-like growth factor-1 and tri-iodothyronine concentrations. A low EA is suspected of suppressing Type 1 immunity [29], and males may respond to LEA by developing an exercise hypogonadal male condition (EHMC), which affects reproductive function. During EHMC, the hypothalamic–pituitary–gonadal axis is disturbed, along with reduced serum testosterone levels (TES). Although TES values remain at the low end of the normal clinical range, symptoms of hypogonadism, such as fatigue, sexual dysfunction, and reduced BMD, are present. The syndrome was first diagnosed in endurance-trained males, although it is also seen in athletes of other disciplines, such as bodybuilding and power disciplines. LEA following restricted dietary EI or EEE can also induce metabolic adaptions to conserve energy, including a decline in the basal metabolic rate (BMR), non-exercise activity thermogenesis (NEAT), and the thermic effect of food (TEF) if the caloric intake is restricted [36].

In this study, the elite male group presented the highest median protein intake among the male groups (1.45 (g/kg/BW). Among the female groups, the highest median value was found in the advanced group (1.42 g/kg/BW). The lowest mean intakes among the female and male groups were found for the intermediate female group and the advanced male group, respectively, which were 1.34 g/kg/BW and 1.25 g/kg/BW. The current recommendation for sport climbers is a daily protein intake of 1.3–1.7 g/kg/BW [3] or as much as 2 g/kg/BW for bouldering [4], which suggests that, considering the median values, with the exception of the male advanced group, other groups have fulfilled the minimum protein intake necessary for the performance. A similar protein intake was observed among Polish advanced climbers by Sas-Nowosielski et al. [11], with a mean intake of 1.48 ± 0.34 g/kg/BW and a higher intake in males (1.59 ± 0.40 g/kg/BW) than females (1.34 ± 0.17). These values are consistent with the previous studies of Monedero et al. [13] of 1.5 ± 0.4 g/kg/BW and Gibson-Smith et al. [12] (1.6 ± 0.5 g/kg/d). Higher intakes were also reported in our previous study in advanced males (1.74 ± 0.6 g/kg/BW), which were lower in the advanced females (1.3 ± 0.42). Furthermore, the protein percentage in the diet ranged from 13% of EI [16] to 17.0 ± 4.2% [14].

Michael et al. [28] demonstrated that 77% of adolescent climbers met or exceeded dietary protein needs, whereas Simič et al. [7] showed that 44% of participants were above the target for protein, with a mean intake of 1.3 ± 0.4 g/kg/BW [7]

Although a higher dietary protein intake may be considered necessary during periods of strength training, no studies have evaluated the protein needs of sports climbers [3].

The recommended CHO intake ranges from 3 to 7 g/kg/BW [3]. In this study, the median CHO intake was the highest in the intermediate group (4.58 g/kg/BW). Similar median values were reported in advanced and elite groups (4.16; 4.22). A lowering trend occurred in the male groups, with the mean CHO intake decreasing with each climbing level. As such, the elite male group had the lowest CHO intake (3.08 g/kg/BW). Monedero et al. [13] (3.6 ± 1.0 g/kg/BW) and Gibson-Smith et al. [12] (3.7 ± 0.9 g/kg/BW) found similar values, and a higher CHO intake was reported in our previous study [15] in the male (4.7 ± 1.4 g/kg/BW) and female (4.6 ±1.3 g/kg/BW) advanced groups. Sas-Nowosielski et al. [11] reported a mean CHO intake of approximately 4 g/kg/BW and indicated that female participants were more likely to cut out CHOs. In Kemmler et al. [16], 54% of the EI was covered by CHOs. Meanwhile, Michael et al. [3] showed that the majority of adolescent climbers (86%) failed to meet the CHO dietary target. In Simič et al. [7], a mean daily CHO intake of 4.3 ± 1.3 g/kg/BW placed 75% of participants below the target value [7].

Some works suggest that a low CHO intake may be connected to the rising popularity of ketogenic diets due to the suggestion that they benefit the reduction in body mass [13]. In addition, a short-duration low-CHO diet, followed by a high-CHO diet, is reported to be a successful nutritional tool for increasing endurance in elite climbers [37]. However, a permanently low CHO intake may be a limiting factor in intensive training [4].

There is no specific recommendation for fat, although it is suggested that the intake should follow the recommendation of the general population and not exceed 35% of the total calorie intake [3]. The fat intake in the diets of participants of this study shifted between median intakes of 1.10 and 1.26 g/kg/BW. The lowest median intake was found in the advanced male group (1.10 g/kg/BW) and the highest intake was found in the elite male group (1.26/kg/BW). Other studies involving sport climbers at different ability levels reported a fat intake of 1.5 ± 0.5 g/kg/BW [13], 1.4 ± 0.4 g/kg/BW [12], and 1.22 ± 0.36 g/kg/BW [11], and 29% of EI [16]. Michael et al. [28] found that 73% of adolescent climbers failed to meet the fat requirements. Furthermore, the mean daily fat intake was 31.8 ± 4.7% of EI, but 37% of participants did not meet the target of 30% EI [7]. In the current study, the elite female group consumed adequate amounts of most micronutrients, except vitamin D and iodine. Meanwhile, the elite male group took in inadequate vitamin D, iodine, and potassium. It is difficult to meet the recommended 15 mcg of vitamin D intake from the diet, and vitamin D supplementation is recommended for the Polish population during all seasons because of insufficient sun exposure [38]. However, no more than 33% of climbers declared using vitamin D supplements [20]. Nonetheless, the available data show that, although the correction of vitamin D deficiency is necessary, additional supplementation may not have added health benefits [39]. The use of salt was not noted in the food diaries. In Poland, regulation dictates that salt producers add iodine [40], meaning that the use of salt in everyday food preparation can significantly increase its intake. Low potassium intake in the male group could be due to low fruit and vegetable intake, as these are its’ greatest sources [24]. Zapf et al. [18] also reported a low vegetable intake in sports climbers.

Monedero et al. [13] demonstrated inadequate calcium, magnesium, and vitamin D intakes in female climbers. Female subjects in the current study had lower than recommended intakes of protein and iron. In comparison to Monedero et al. [13], the males and females in this study consumed more calcium, magnesium, iron, and vitamin C, and similar amounts of vitamin D and B12. The elite female climbers had the highest iron, vitamin C, potassium, magnesium, and zinc intakes. However, these findings contrast with previous work showing deficiencies in the diets of elite female sports climbers [12].

Compared to the study by Kemmler et al. [16], male climbers presented lower intakes of calcium, magnesium, and potassium, similar intakes of iron, vitamins B, and E, and higher intakes of vitamins A and D. In this study, among the male groups, the elite climbers presented the lowest intakes of micronutrients compared to the other climbing levels. However, their diets still fulfilled most of the EAR recommendations.

The observation that the elite female climbers had the highest intakes of most micronutrients, despite the lowest EA value, suggests that this group puts great focus on the quality of their diet. In the elite male group, taking into account their low EI, the micronutrient intake was satisfactory. However, as a higher training volume would increase their need for many vitamins and minerals, more emphasis should be placed on choosing foods of sufficient quality.

## 5. Study Limitations

Despite the best efforts to ensure that the examination was of the best quality, there were some limitations that should be mentioned.

Exercise energy expenditure was set based on the declared hours of training using the MET logs for the activity type. The exact time during which training was performed should be measured using more specialized tools, allowing the collection of more reliable data in terms of time and the specifics of the activity, which differ in energy requirements, such as warm ups, different training types, and stretching. However, some of the previous studies mentioned in the discussion also used METs, based upon which our results are comparable.

Another limitation which should be considered is that our study involved climbers of different climbing types, namely bouldering and rope climbing, with each of these types having slightly different needs in terms of the energy substrates used for the training efforts, which influences the general macronutrient intake of participants representing each climbing type. However, Olympic-style climbing includes both of these types, and many climbers do not focus on one type of climbing, which is why such a specification in the study group significantly limits the amount of participants.

## 6. Conclusions

The significant differences in EAs between the male groups, with both the male and female elite groups presenting the lowest values, suggest that the athletes of this discipline are prone to energy undernutrition regardless of gender. However, the differences between the dietary intake of micronutrients between various levels of advancement were more visible in the female groups, with the elite group having the highest intake of most micronutrients. In the male groups, the consumption of micronutrients was low among all ability levels, but it still reached the recommended values.

In the elite group, a high-quality diet was observed despite very low energy availability. Despite the appropriate supply of micronutrients, a long-term insufficient intake of energy can lead not only to a limitation in training opportunities but also to hormonal disorders in both women and men. The low supply of carbohydrates and high supply of fat and protein in the group of elite men may suggest the use of fashionable low-carbohydrate diets, the effectiveness of which has not been confirmed in the case of athletes.

It is necessary to educate sports climbers about the importance of proper nutrition and the consequences of insufficient EI.

## Figures and Tables

**Table 1 ijerph-20-05176-t001:** Characteristics of study groups.

Measurement	Female			Kruskal–Wallis Test		Male			Kruskal–Wallis Test	
	Intermediate (n = 15)	Advanced (n = 20)	Elite (n = 5)	H	*p*	Intermediate (n = 32)	Advanced (n = 29)	Elite (n = 5)	H	*p*
	M (Q1–Q3)	M (Q1–Q3)	M (Q1–Q3)			M (Q1–Q3)	M (Q1–Q3)	M (Q1–Q3)		
Age (y)	31.0 (26.0–36.0)	28.5 (24.0–34.0)	29.0 (25.0–30.0)	0.93	0.63	31.0 (26.5–34.5)	31.0 (28.0–33.0)	30.0 (28.0–30.0)	0.63	0.73
Body Weight (kg)	58.7 (57.2–62.3)	55.6 (49.1–60.8)	58.2 (55.2–59.8)	2.42	0.30	72.8 (66.6–80.0)	70.6 (66.5–75.9)	66.5 (64.3–69.9)	3.83	0.15
Height (cm)	167.0 (166.0–170.0)	164.8 (161.3–167.8)	170.0 (161.0–172.0)	2.38	0.30	177.5 (173.8–183.0)	178.0 (173.0–181.5)	177.0 (172.0–180.0)	0.37	0.83
Lean Body Mass (kg)	44.9 (42.9–49.2)	43.2 (39.8–47.4)	45.4 (42.5–46.7)	2.04	0.36	61.2 (55.9–64.2)	58.6 (55.4–63.3)	56.0 (54.0–58.7)	1.85	0.40
Body Fat (%)	22.6 (17.6–25.6)	22.3 (18.5–24.8)	21.9 (19.3–22.0)	0.86	0.65	17.6 (14.7–20.0)	15.9 (13.8–18.3)	16.0 (15.8–16.0)	2.96	0.23
Body Fat (kg)	14.1 (12.3–15.5)	12.4 (8.6–15.1)	12.7 (11–12.8)	3.22	0.20	13.4 (9.7–14.9)	11.1 (8.7–13.1)	10.3 (9.5–10.5)	5.66	0.06
BMI (kg/m^2^)	21.0 (20.3–22.1)	20.6 (19.0–21.8)	20.1 (20.0–20.2)	2.24	0.33	23.2 (21.7–24.8)	22.5 (21.7–23.5)	21.2 (21–21.6) #	6.84	0.03
Grip Strength-to-Body Mass Ratio	0.6 (0.5–0.6)	0.5 (0.5–0.6)	0.6 (0.5–0.7)	1.33	0.51	0.6 (0.6–0.7)	0.7 (0.6–0.7)	0.8 (0.8–0.9) &,#	14.37	0.00
Contraction Force Right (kg)	35.6 (26.3–37.0)	31.4 (29.4–34.9)	35.5 (31.2–37.3)	1.79	0.41	47.5 (44.0–52.6)	50.4 (43.4–54.8)	55.3 (53.3–59.6)	4.27	0.12
Contraction Force Left (kg)	32.0 (25.1–34.7)	27.8 (25.6–31.2)	29.0 (27.7–35.0)	1.88	0.39	43.8 (39.9–48.7)	45.1 (41.2–51.9)	51.8 (51.8–52.2) #	6.39	0.04
Systolic Blood Pressure	107.0 (101.0–109.0)	106.0 (100.5–113.0)	92.0 (91–92) &,#	8.48	0.01	122.5 (113–127.5)	120.0 (11.0–131.0)	118.0 (114.0–126.0)	0.38	0.83
Diastolic Blood Pressure	70.0 (66.0–75)	68.0 (60.5–73.5)	54.0 (52.0–64.0) #	6.74	0.03	74.0 (69.0–79.5)	72.0 (66.0–78.0)	73.0 (69.0–76.0)	1.22	0.54
Resting Heart Rate (bpa)	67.65.0 (62.0–72.0)	61.5 (58–71)	57.0 (55.0–61.0)	4.41	0.11	64.0 (57.0–72.5)	60.0 (54.0–68.0)	62.0 (47.0–72.0)	1.98	0.37
RMR	1527 (1419–1694)	1534.5 (1416.5–1635.0)	1616.0 (1556.0–1653.0)	1.64	0.44	1954.5 (1889.0–2066.0)	1990.0 (1819.0–2129.0)	1873.0 (1872.0–2125.0)	0.17	0.92

M—median; Q1— first quartile; Q3—third quartile; &—significant differences (*p* < 0.05) in relation to advanced group; #—significant differences (*p* < 0.05) in relation to intermediate group.

**Table 2 ijerph-20-05176-t002:** Comparison of energy availability (kcal/kg FFM/day) between different climbing levels in the male and female groups and between sexes, and the proportion of each cohort (%) with low EA (<30 kcal/kg/FFM/day).

Female							Kruskal–Wallis Test		Mann–Whitney U Test	
							H	*p*	Z	*p*
Whole cohort (n = 40)	Intermediate (n = 15)	Proportion of cohort with EA below suboptimal (%)	Advanced (n = 20)	Proportion of cohort with EA below suboptimal (%)	Elite (n = 5)	Proportion of cohort with EA below suboptimal (%)	4.23	0.10	2.04	0.04
M (Q1–Q3)	M (Q1–Q3)		M (Q1–Q3)		M (Q1–Q3)					
31.56 (21.17–37.57) x	33.4 (28.1–41.0)	27	27.6 (20.9–36.6)	55	13.4 (10.6–33.7)	60				
**Male**							11.42	0.00		
Whole cohort (n = 66)	Intermediate (n = 32)	Proportion of cohort with EA below suboptimal (%)	Advanced (n = 29)	Proportion of cohort with EA below suboptimal (%)	Elite (n = 5)	Proportion of cohort with EA below suboptimal (%)				
M (Q1–Q3)	M (Q1–Q3)		M (Q1–Q3)		M (Q1–Q3)					
24.74 (18.30–33.18)	28.5 (24.7–34.3)	53	20.9 (15.4–25.6) *	79	18.9 (10.5–22.5) *	100				

M—median; Q1—first quartile; Q3—third quartile; x—significant difference (*p* < 0.05) in relation to male, *—significant differences (*p* < 0.05) in relation to intermediate group.

**Table 3 ijerph-20-05176-t003:** Comparison of macronutrient intake per kilogram of body weight between different climbing levels and between genders.

Macronutrient	Female				Male				Kruskal–Wallis Test		Mann–Whitney U Test	
	Whole cohort (n = 40)	Intermediate (n = 15)	Advanced (n = 20)	Elite (n = 5)	Whole Cohort (n = 66)	Intermediate (n = 32)	Advanced (n = 29)	Elite (n = 5)	H	*p*	Z	*p*
	M (Q1–Q3)	M (Q1–Q3)	M (Q1–Q3)	M (Q1–Q3)	M (Q1–Q3)	M (Q1–Q3)	M (Q1–Q3)	M (Q1–Q3)				
Protein (g/kg)	1.36 (1.22–1.55)	1.34 (1.23–1.47)	1.42 (1.16–1.61)	1.40 (1.20–1.86)	1.38 (1.10–1.63)	1.42 (1.16–1.70)	1.25 (1.03–1.55)	1.45 (1.34–1.46)	0.89	0.64	–0.13	0.90
Carbohydrate (g/kg)	4.41 (3.74–5.38) x	4.58 (4.00–5.39)	4.16 (3.42–5.27)	4.22 (4.20–5.79)	3.75 (2.92–4.76)	3.95 (3.23–4.80)	3.79 (2.84–4.51)	3.08 (2.91–3.64)	0.67	0.71	–2.78	0.01
Fat (g/kg)	1.57 (0.92–1.41)	1.22 (0.95–1.60)	1.12 (0.92–1.33)	1.20 (0.84–1.23)	1.18 (0.98–1.43)	1.22 (1.01–1.43)	1.10 (0.89–1.43)	1.26 (1.05–1.37)	1.22	0.54	0.15	0.87

M—median; Q1—first quartile; Q3—third quartile; x—significant difference (*p* < 0.05) in relation to Male.

**Table 4 ijerph-20-05176-t004:** Comparison of the macronutrient and micronutrient intakes of female and male sports climbers of different climbing levels.

**Female**						
**Component**		**Intermediate (n = 15)**	**Advanced (n = 20)**	**Elite (n = 5)**	**Kruskal–Wallis Test**	
		**M (Q1–Q3)**	**M (Q1–Q3)**	**M (Q1–Q3)**	**H**	** *p* **
Calories (kcal)		2046.2 (1782.6–2287.0)	1721.4 (1505.0–2080.4)	1684.8 (1635.3–2550.0)	4.27	0.11
Protein (g)		79.5 (71.8–85.2)	76.5 (66.1–90.8)	81.6 (72.5–96.8)	0.54	0.76
Carbohydrates (g)		282.3 (226.4–325.9)	228.3 (199.4–268.4)	253.1 (212.0–346.1)	3.09	0.21
Total Dietary Fiber (g)		31.1 (25.00–36.1)	25.0 (19.8–28.2)	39.8 (39.5–39.8) &	10.99	0.00
Fat (g)		70.4 (57.3–93.5)	59.9 (53.7–78.0)	68.1 (50.7–69.7)	3.29	0.19
Saturated Fat (g)		24.8 (19.3–27.9)	20.0 (16.3–23.2)	21.4 (30.7–24.3)	3.93	0.14
Monosaturated Fat (g)		24.8 (19.2–32.4)	22.3 (18.4–32.2)	26.5 (16.5–29.0)	0.72	0.70
Polysaturated Fat (g)		12.6 (10.9–18.6)	12.3 (8.0–14.8)	9.6 (8.5–14.4)	0.68	0.71
Cholesterol (mg)		301.9 (211.3–457.3)	268.9 (206.0–346.8)	280.3 (179.2–365.2)	0.44	0.82
EAR						
Vitamin A (mcg)	500	1290.5 (896.7–1851.2)	1061.2 (766.3–1581.5)	1893.9 (1220.7–3405.4)	4.05	0.13
Vitamin B1 (mg)	0.9	1.38–0.98–1.47)	1.10 (0.89–1.51)	1.47 (1.23–1.53)	1.90	0.39
Vitamin B2 (mg)	0.9	1.75 (1.45–1.98)	1.58 (1.31–1.97)	1.74 (1.66–2.06)	0.93	0.63
Vitamin B3 (mg)	11	17.4 (14.4–21.0)	16.9 (11.1–23.6)	18.7 (16.8–18.9)	0.18	0.91
Vitamin B6 (mg)	1.1	1.97 (1.78–2.32)	1.93 (1.57–2.48)	2.65 (2.24–3.32)	4.53	0.10
Vitamin B12 (mcg)	2	2.89 (2.08–4.88)	4.31 (2.235–6.535)	2.33 (1.33–4.16)	2.24	0.33
Vitamin C (mg)	60	158.2 (110.9–212.7)	143.5 (85.8–180.6)	219.2(171.1–254.7)	3.84	0.14
Vitamin D (mcg) ^AI^	15	2.5 (1.3–5.9)	3.9 (1.3–7.2)	2.66 (0.94–4.06)	0.71	0.70
Vitamin E (mg) ^AI^	8	12.5 (8.6–15.3)	10.6 (7.1–14.2)	11.49 (10.9–16.9)	1.67	0.44
Folate (mcg)	320	453.1 (370.0–502.3)	329.0 (300.5–490.6)	469.4 (445.7–518.9)	4.13	0.13
Calcium (mg)	800	705.7 (536.7–1096.7)	703.1 (604.96–817.3)	856.7 (752.8–1107.1)	2.86	0.24
Iodine (mcg)	95	43.0 (37.2–65.5)	28.8 (15.8–36.6) *	41.6 (21.3–46.0)	8.39	0.02
Iron (mg)	6	16.5 (13.9–17.4)	12.4 (11.1–15.5)	19.7 (17.8–20.6) &	12.76	0.00
Magnesium (mg)	255/265 ^a^	383.1 (322.1–505.3)	317.2 (245.4–356.6)	472.5 (410.6–509.6) &	11.40	0.00
Phosphorus (mg)	580	1369.8 (1191.8–1545.5)	1203.0 (1043.5–1450.7)	1535.4 (1400.0–1739.0) &	7.05	0.03
Potassium (mg) ^AI^	3500	3365.1 (2824.2–3925.0)	2977.3 (2503.2–3484.1)	3921.9 (3863.6–4431.7) &	6.11	0.05
Sodium (mg) ^AI^	1500	2325.6 (1518.1–2189.5)	1909.6 (1339.1–2447.7)	2672.0 (2636.6–2767.1)	3.28	0.19
Zinc (mg)	8	11.3 (10.3–12.3)	9.5 (7.1–11.5)	11.7 (11.5–12.5) &	6.36	0.04
**Male**						
**Component**		**Intermediate (n = 32)**	**Advanced (n = 32)**	**Elite (n = 5)**	**Kruskal–Wallis Test**	
		**M (Q1–Q3)**	**M (Q1–Q3)**	**M (Q1–Q3)**	**H**	** *p* **
Calories (kcal)		2391.2 (2020.1–2660.1)	2080.2 (1798.6–2594.8)	2010,9 (1967.2–2058.8)	4.15	0.13
Protein (g)		107.7 (85.2–118.0)	87.6 (74.9–109.7)	96.5 (86.0–101.4)	3.33	1.19
Carbohydrates (g)		288.1 (230.9–335.6)	265.9 (200.1–326.2)	222.4 (179.8–254.4)	3.80	0.15
Total Dietary Fiber (g)		28.4 (23.1–38.7)	25.8 (18.1–31.8)	20.0 (18.0–28.7)	2.38	0.30
Fat (g)		86.9 (73.7–103.8)	79.1 (66.8–98.3)	88.0 (69.8–90.7)	1.15	0.56
Saturated Fat (g)		30.2 (24.4–38.7)	25.4 (21.2–34.4)	25.1 (18.1–34.3)	2.59	0.27
Monosaturated Fat (g)		31.7 (26.1–41.2)	28.2 (24.4–34.9)	33.9 (30.2–35.1)	1.91	0.39
Polysaturated Fat (g)		14.3 (9.8–19.9)	16.5 (10.5–21.8)	14.3 (14.1–14.6)	1.25	0.54
Cholesterol (mg)		440.2 (248.8–710.7)	403.7 (248.3–527.8)	445.0 (337.5–666.2)	0.89	0.64
EAR						
Vitamin A (mcg)	630	1231.3 (853.5–1836.6)	1404.1 (750.1–1535.0	717.2 (708.2–875.7)	4.31	0.11
Vitamin B1 (mg)	1.1	1.44 (1.31–1.71)	1.29 (1.05–1.64)	1.15 (1.10–1.17)	5.23	0.07
Vitamin B2 (mg)	1.1	2.11 (1.78–2.48)	1.85 (1.50–2.39)	1.79 (1.65–1.98)	2.88	0.24
Vitamin B3 (mg)	12	20.5 (17.2–28.2)	16.4 (13.8–26.4)	18.5 (17.7–21.7)	3.35	0.19
Vitamin B6 (mg)	1.1	2.59 (2.12–2.96)	1.93 (1.57–2.90) *	1.78 (1.71–2.03)	7.84	0.02
Vitamin B12 (mcg)	2	4.39 (3.07–7.57)	3.31 (2.30–5.27)	3.31 (2.9–3.54)	3.36	0.19
Vitamin C (mg)	75	155.8 (93.6–193.9)	125.3 (76.9–167.1)	101.8 (101.0–166.1)	1.89	0.39
Vitamin D (mcg) ^AI^	15	3.73 (2.20–6.58)	2.42 (1.61–3.82)	3.49 (3.18–3.80)	3.51	0.17
Vitamin E (mg) ^AI^	10	12.8 (10.1–16.7)	11.2 (8.0–17.5)	10.1 (9.1–11.9)	1.29	0.52
Folate (mcg)	320	434.8 (364.4–548.8)	426.0 (354.9–561.9)	348.2 (308.0–443.9)	1.30	0.52
Calcium (mg)	800	867.9 (721.1–1121.4)	773.4 (629.6–1112.0)	874.7 (791.3–997.7)	1.93	0.38
Iodine (mcg)	95	44.9 (29.9–63.1)	29.0 (24.0–48.4)	37.5 (26.5–38.1)	5.02	0.08
Iron (mg)	6	17.0 (14.2–19.3)	13.6 (12.4–15.8) *	12.9 (12.9–13.4)	7.37	0.03
Magnesium (mg)	330/355 ^a^	412.8 (324.8–496.9)	356.5 (292.1–472.0)	325.8 (283.0–362.6)	3.08	0.21
Phosphorus (mg)	580	1754.6 (1482.2–1991.3)	1487.7 (1176.6–1651.7) *	1533.1 (1471.1–1567.0)	7.84	0.02
Potassium (mg) ^AI^	3500	3785.1 (3050.1–4335.2)	3316.3 (2522.2–3786.1)	2653.0 (2638.8–3093.7)	5.81	0.05
Sodium (mg) ^AI^	1500	2604.4 (1631.5–3157.5)	2182.6 (1664.4–2967.7)	1438.6 (711.3–2076.1)	4.05	0.13
Zinc (mg)	11	13.3 (12.3–14.5)	11.0 (9.1–12.7) *	11.0 (10.7–11.3)	12.43	0.00

M—median; Q1—first quartile; Q3—third quartile; EAR—estimated average requirement; AI—adequate intake; ^a^—recommendation for 19–30 y/recommendation for 31–59 y; &—significant differences (*p* < 0.05) in relation to advanced group, *—significant differences (*p* < 0.05) in relation to intermediate group.

## Data Availability

The data presented in this study are available upon request from the corresponding author. The data are not publicly available due to privacy restrictions.

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
