# Peer review of "The Evaluation of Energy Availability and Dietary Nutrient Intake of Sport Climbers at Different Climbing Levels"

_ijerph, 2023, doi:10.3390/ijerph20065176_

Round 1

Author Response

Thank you very much for reading and reviewing my work. It was a pleasure to get feedback about the manuscript submitted. I found all of your comments very inspirational and I’m sure introducing the changes following your suggestions will add value to the manuscript.

Following your suggestion I specified the aim of the study. I wrote about the gender specific. The anthropometric measurements were taken for estabilishing the characteristic of the group. I am writing about the BMI in the first part of discussion, however, apart of the signifficant differences between the male group I pointed in the results section, there was no intention to study BMI among sport climbers.

I also focused on expanding and clarifying the materials and methods section. I added the subheadings: design, procedure, participants, and outcome measures. However it was not an interventional study, and no intervetion was carried out so I didn’t feel empowered to put the “Intervention” part in this section.

I was a bit confused about the suggestion about the subheadings in the results section as there was no subheadings there. Hovewer I tried to put the study outcome measures section, results section and discussion In simillar order, so it was clear.

I deleted subheadings from the discussion section, but as it hasn’t been evaluated yet, I’m not sure if it was the issue mentioned in the comment.

I hope now my work gained more clarity and value and will be worth reading.

Reviewer 2 Report

With regard to the manuscript: The Evaluation of Energy Availability and Dietary Nutrient Intake of Sport Climbers at Different Climbing Levels, submitted to Int. J. Environ. Res. Public Health, 2023. The general scope of the study appears to be acceptable and is of interest, but the novelty of the study could be more highlighted. Nutritional investigations in climbers have been well established, but what is new. The authors would explain why their findings aggregates to the existing knowledge. Allow me to give you a few suggestions.

Introduction

·  The authors do not explain the reason why they decided to explore intermediate, advanced and elite climbers. Why not was emphasized the possible differences between climbing disciplines? (since the authors reported that longer climbing routes emphasize endurance, whereas bouldering relies more on power). Hypotheses must be proposed in the context of climbing levels and taking into consideration gender differences.

·  According to authors: “research has shown inadequate energy availability (EA) [7], unbalanced food quality, and poor nutrient timing in the diets of climbers of different ages and levels of advancement [18].” However, the novelty of the study could be more highlighted (in introduction). The novelty of study would be the comparison between climbing levels, genders?, would be the quantification of micronutrients? It should be clear for the reader.

Methods

·   Evaluations of meals were photographs-based. Could you detail how accurate is this? The food should be weighted. One caveat for the use of photographs is that they cannot capture with precision the real energy intake. I understand the difficult for collecting food intake information in humans, however I strongly believe that these limitations should be assumed. A clear discussion of the limitations of the study is missing and has to be included.

·  (line 126) I suggest the inclusion of a figure that summarize the obtaining of EA. This would be better for didactic purposes since energy exercise expenditure (EEE) and energy intake (EI) are “output and input” for the nutrition field (like a drain and faucet).

·  Give more details about tests using hand grip dynamometer.

·  The statistical procedures is not bad, but there are matters to be improved. Kruskal-Wallis test is a nonparametric version of the ANOVA. The authors should give more details about the normality assumptions.

·  About data collection method, it is important describe WHEN? (Time of day, environmental condition, season of data collection, data collection duration).

·  Researchers were trained or had training in conducting tests such as blood pressure measures, anthropometric measurements…?

Results

·  Error in table1: Contrction force left (kg)

·  (in abstract) “Higher climbing levels correlated with 16 lower EA in the male group”. Revise this phrase since in this study was not performed correlations.

·  Post hoc differences among groups in table 1 must be more detailed. For example: there is a symbol ** (indicating difference between elite 53.04 vs. intermediate 47.18) for contraction force left - male, however why advanced group, which exhibited a force of 45.7, showed no significant difference in relation to elite group. Please, it is an opportunity to carefully revise the statistical data. Perhaps, you are interpreting results only through of H-test values from Kruskal Wallis, but not from post hoc p-values.

·  table 1: pulse (bpa). It might be better to include “resting heart rate” since was not evaluated during exercise conditions.

·  The table 2 could be more harnessed regarding energy availability. Please, try exhibit percentual data in relation to nutritional recommendations /guidelines. What values (for EA) can be considered as optimal in order to ensure caloric intake. Post hoc differences among groups in table 2 must also be revised.

·  Table 3. Mann Whitney U test should be made for each comparison (male and female climbers for intermediate), (male and female climbers for Advanced) and (male and female climbers for Advanced for Elite). Thus, would be coherent to expect three Z- test values (but this was not made).

Discussion

·  It is well written showing interesting. However, a figure (a big picture) summarizing all findings in relation to the literature could be useful.

Author Response

Thank you very much for reading and reviewing my work. It was a pleasure to get feedback about the manuscript submitted. I found all of your comments very inspirational and I’m sure introducing the changes following your suggestions will add value to the manuscript.

Following your suggestion, I added information about the novelty of the study.

I also focused on expanding and clarifying the materials and methods section. I added the subheadings: design, procedure, participants, and outcome measures. I tried to describe the whole procedure with the best accuracy and all details. The meal photographs were used as the additional tool in case of doubts, as the application allowed covert house measures of products to grams, in case the photograph content differed from the meal entered into the app, the participant was contacted by a dietitian to verify the information given. However, I also added the Limitation section in the manuscript to point out aspects that could have been done with better precision or were visible limitations.

I decided to consult the statistic specialist according to the suggestions about the statistics. First, I changed the mean values, which were misleading regarding the outcomes of the statistical results ( as you mentioned in table 1) for the median. As nonparametric tests were used in all cases, median and quartiles should be used to present data- this was changed in all tables, results descriptions, and discussions.

As adding all Z values would make the table overcrowded with the results I decided to put that information, as well as all the post-hoc results in the brackets while describing the results. Hope it will be clear and makes the results transparent. However, I found it super inspirational to add the information about the proportion of cohorts that do not exceed the minimum value of suboptimal EA- I added this information to the table and I think it puts a new light on the data presented. Thanks a lot.

I was considering the suggestion about the table summarising all the findings according to the works I referred to in the discussion, but the variety of topics, discussed in the manuscript would make it hard to show it clearly. It would be a perfect way to focus on such a summary as a topic of the review work- that I would do with pleasure as another project.

I hope you find my answers satisfying, and the improvement of my work after your suggestions will make it a valuable manuscript.

Author Response

Thank you very much for reading and reviewing my work. It was a pleasure to get feedback about the manuscript submitted. I found all of your comments very inspirational and I’m sure introducing the changes following your suggestions will add value to the manuscript.

Following your suggestion, I changed the keywords and added statistical information in the abstract.

I focused on the correction of methodology adding a lot of information on how the study was conducted.

I rewrote indicated sentences, and added references where it was necessary.

In the tables, I show statistic importance with p-value, and H test value or Z test value depending on the test used. I also added the posthoc test results in the sections where describing data in the tables.

I checked the reference list.

I believe those changes will be satisfactory, to accept the manuscript.

Round 2

Author Response

Thank you very much for reviewing my work again and for pointing out inaccuracies in the presentation of statistical results.

I changed all the tables following your suggestions ( I did not do it in the “track changes” function for as I changed the whole tables and instered new ones):

  • I inserted signs of significance next to the data to which they refer to
  • I added the median results of whole cohort of female and males in the tables in which I present the Mann Whitney U test between the genders
  • I corrected the tables with nutrients- thank you for pointing out the visible mistake- I decribed the results in male group ( differences were significant between Intermediate vs Advanced) but the information in the table where not convergent with the text.
  • I revised the results once again to make sure there are no more such mistakes.

I hope you find my answers satisfying, and the improving of my work after your suggestions will make it a valuable manuscript.